# AlphaFold Meets Flow Matching
# for Generating Protein Ensembles

**Bowen Jing, Bonnie Berger, Tommi Jaakkola**
CSAIL, Massachusetts Institute of Technology
bjing@mit.edu, {bab, tommi}@csail.mit.edu

## Abstract

Recent breakthroughs in protein structure prediction have pointed to structural en-
sembles as the next frontier in the computational understanding of protein structure.
At the same time, iterative refinement techniques such as diffusion have driven
significant advancements in generative modeling. We explore the synergy of these
developments by combining AlphaFold and ESMFold with flow matching, a power-
ful modern generative modeling framework, in order to sample the conformational
landscape of proteins. When trained on the PDB and evaluated on proteins with
multiple recent structures, our method produces ensembles with similar precision
and greater diversity compared to MSA subsampling. When further fine-tuned on
coarse-grained molecular dynamics trajectories, our model generalizes to unseen
proteins and accurately predicts conformational flexbility, captures the joint dis-
tribution of atomic positions, and models higher-order physiochemical properties
such as intermittent contacts and solvent exposure. These results open exciting
avenues in the computational prediction of conformational flexibility.

## 1 Introduction

The success of AlphaFold2 [12] at the 14th Critical Assessment of Structure Prediction (CASP)
marked a watershed moment in the computational understanding of protein structure. Since then,
predicting multiple conformational states of proteins has emerged as the next frontier towards the
ultimate aim of understanding protein *function* from sequence [13, 17]. Significant conformational
changes are critical in the function of transporters, channels, enzymes, motor proteins, receptors,
and many other proteins, and even relatively static proteins experience thermal fluctuations whose
properties are implicated in the strength and selectivity of molecular interactions. Hence, a generative
model of protein structure which builds upon the level of accuracy of single-structure predictors, but
reveals conformational heterogeneity, would be of great value to computational structure biology.

In this work, we combine highly accurate protein structure prediction models such as AlphaFold2 [12]
and ESMFold [14] with *flow matching* [15, 4], a powerful modern generative modeling framework,
in order to sample the conformational landscape of proteins. Structure prediction models have been
trained as *regression* models based which predict a single protein structure for a given (sequence
or MSA) input. Our synthesis relies on the key insight that modern iterative generative modeling
frameworks such as diffusion and flow-matching provide a general recipe to convert such regression
models to become *generative models* of protein structure with relatively little modification to the
architecture and training objective. Thus, by fine-tuning structure prediction models on the generative
modeling objectives, we obtain powerful and accurate distributional models of protein structure. In
particular, while existing approaches to generating structural ensembles have focused on modifying
the MSA input via subsampling [8], mutagenesis [20], or clustering [24], our method generates
ensembles without resorting to such input ablation techniques and thus can be equally applied to
language-model based structure predictors such as ESMFold.

NeurIPS 2023 AI for Science Workshop.

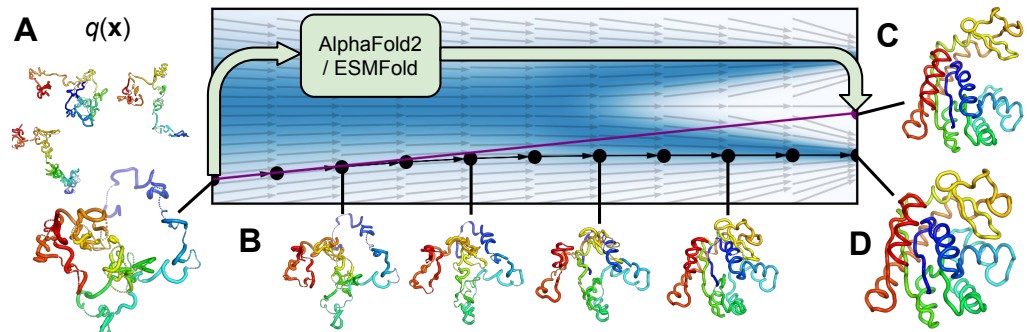

Figure 1: **Conceptual overview of AlphaFlow / ESMFlow.** (A) Samples are drawn from a harmonic (polymer-structured) prior. (B) The structure is progressively refined under a neural ODE (C) At each step, the AlphaFold2 or ESMFlow prediction parameterizes the direction of the flow. (D) The final structure is a sample from the predicted distribution of structures.

We investigate the performance of our flow-matching variants of AlphaFold2 and ESMFold (named AlphaFlow and ESMFlow) in two settings. First, we aim to predict diverse yet accurate ensembles for recently deposited proteins in the PDB. Second, we exploit the ability to train on ensembles beyond the PDB to model conformational ensembles from coarse-grained molecular dynamics. In both settings, our model successfully generates a diverse set of structures and in the latter case accurately recapitulates physiochemical properties of the ground truth ensembles.

## 2 Method

### 2.1 Flow Matching

*Flow matching* [15, 4, 3] is a recently proposed generative modeling paradigm that resembles and builds upon the significant success of diffusion models [10, 19] in generative modeling of images and molecules. The fundamental object in flow matching is a conditional probability path $p_t(\mathbf{x} \mid \mathbf{x}_1), t \in [0, 1]$: a family of densities conditioned on a data point $\mathbf{x}_1 \sim p_{\text{data}}$ which interpolates between a common prior distribution $p_0(\mathbf{x} \mid \mathbf{x}_1) = q(\mathbf{x})$ and an approximate Dirac $p_1(\mathbf{x} \mid \mathbf{x}_1) \approx \delta(\mathbf{x} - \mathbf{x}_1)$. Given a conditional vector field $u_t(\mathbf{x} \mid \mathbf{x}_1)$ that generates the time evolution of this conditional probability path, one then regresses against the *marginal vector field* with a neural network:

$$\hat{v}(\mathbf{x}, t; \theta) \approx v(\mathbf{x}, t) := \mathbb{E}_{\mathbf{x}_1 \sim p_t(\mathbf{x}_1 \mid \mathbf{x})}[u_t(\mathbf{x} \mid \mathbf{x}_1)] \tag{1}$$

At convergence, the learned vector field $\hat{v}(\mathbf{x}, t; \theta)$ is a neural ODE that evolves the prior distribution $q(\mathbf{x})$ to the data distribution $p_{\text{data}}(\mathbf{x})$ [15]. Score-matching in diffusion models can be seen as a special case of flow matching; however, as discussed in Appendix A.1, flow matching for protein structure circumvents many difficulties that would otherwise arise with diffusion.

Designing a flow-matching generative framework amounts to the choice of a conditional probability path and its corresponding vector field. Inspired by the interpolant-based perspective on flow matching [4], we define the conditional probability paths via the sampling process

$$\mathbf{x} \mid \mathbf{x}_1 = (1 - t) \cdot \mathbf{x}_0 + t \cdot \mathbf{x}_1, \quad \mathbf{x}_0 \sim q(\mathbf{x}_0) \tag{2}$$

i.e., by sampling a noise from the prior $q(\mathbf{x})$ and interpolating linearly with the data point. This conditional probability path is associated with the conditional flow field

$$u_t(\mathbf{x} \mid \mathbf{x}_1) = (\mathbf{x}_1 - \mathbf{x})/(1 - t) \tag{3}$$

which resembles the field proposed in [5] with $\kappa(t) = 1 - t$. Customarily, we then learn a neural network to approximate the marginal vector field according to Equation 1. However, a reparameterization $\hat{v}(\mathbf{x}, t; \theta) = (\hat{\mathbf{x}}_1(\mathbf{x}, t; \theta) - \mathbf{x})/(1 - t)$ reveals that we can equivalently learn the expectation of $\mathbf{x}_1$:

$$\hat{\mathbf{x}}_1(\mathbf{x}, t; \theta) \approx \mathbb{E}_{\mathbf{x}_1 \sim p_t(\mathbf{x}_1 \mid \mathbf{x})}[\mathbf{x}_1] \tag{4}$$

This reparameterization is very similar to those employed for image diffusion models [10]. Crucially, since $\mathbf{x}_1$ refers to samples from the data distribution, this allows the use of models that predict the data itself, i.e., *protein structures*, to be easily adapted in a flow-matching framework.

## 2.2   Flow Matching for Protein Ensembles

To apply flow matching to protein ensembles, we consider a protein structure to be described by the 3D coordinates of its $\beta$-carbons ($\alpha$-carbon for glycine): $\mathbf{x} \in \mathbb{R}^{N \times 3}$. We then choose the prior distribution $q(\mathbf{x})$ over the positions of these $\beta$-carbons to be a *harmonic prior* [11] defined by

$$q(\mathbf{x}) \propto \exp \left[ -\frac{\alpha}{2} \sum_{i=1}^{N-1} \|\mathbf{x}_i - \mathbf{x}_{i+1}\|^2 \right] \qquad (5)$$

This prior distribution ensures that the samples along the conditional probability path, and hence inputs to the neural network $\hat{\mathbf{x}}_1$, always remain polymer-like, physically plausible 3D structures. To parameterize the neural network $\hat{\mathbf{x}}_1(\cdot, t; \theta)$, we can directly use the AlphaFold2 and ESMFold protein structure prediction models if we modify them to accept an *input structure* $\mathbf{x}$ corresponding to the state at time $t$, in addition to the protein sequence or MSA. Not coincidentally, this is reminiscent of the idea of *template structures* employed by certain variants of AlphaFold2. Thus, we develop an input embedding module very similar to AlphaFold2's template embedding stack (detailed in Appendix A) and fine-tune both AlphaFold2 and ESMFold with this additional module and input.

The parameterization of learning the conditional expectation of $\mathbf{x}_1$ (Equation 4) suggests that the neural network should be trained with an $L_2$-loss, i.e., MSE. However, there are several issues with this direct approach. (1) The structure prediction networks not only predict $\beta$-carbon coordinates, but also all-atom coordinates and residue frames. (2) The input to the network is $SE(3)$-invariant by design, and thus the network can distinguish outputs only up to the action of $SE(3)$, which would occur arbitrarily high MSE loss. (3) The networks obtain best performance (and were orginally trained) with the $SE(3)$-invariant Frame Aligned Point Error (FAPE) loss.

To reconcile these issues with the flow-matching framework, we redefine the space of protein structures to be the *quotient* space $\mathbb{R}^{3 \times N} / SE(3)$, with the prior distribution projected to this space. We redefine the interpolation between two points in this space to be linear interpolation in $\mathbb{R}^3$ after RMSD-alignment. Further, because the quotient space is no longer a vector space, there is no longer a notion of "expectation" of a distribution; instead, we aim to learn the more general Fréchet mean of the conditional distribution $p(\mathbf{x}_1 \mid \mathbf{x})$:

$$\hat{\mathbf{x}}_1(\mathbf{x}, t; \theta) \approx \min_{\hat{\mathbf{x}}_1} \int \mathrm{FAPE}^2(\mathbf{x}_1, \hat{\mathbf{x}}_1) \, dp_t(\mathbf{x}_1 \mid \mathbf{x}) \qquad (6)$$

where we have leveraged the convenient property that FAPE is a valid metric [12]. This regression target gives rise to a training loss identical to the original FAPE, except now *squared*. The squaring is in fact essential to the ability to model multiple conformations: the original AlphaFold2 preferentially models a single conformation rather than a mixture of multiple correct conformations, which is a direct consequence of the triangle inequality respected by FAPE. With these modifications, the final result for the training and inference procedures are detailed in Appendix A. While the changes are technical deviations from the direct flow matching framework, they are appropriate adaptations to the problem domain and we find them to work well in practice.

## 3   Experiments

We fine-tune both AlphaFold2 and ESMFold on the PDB with our flow-matching framework. We use the OpenFold [2] implementation of AlphaFold2 and the original CASP14 weights, as well as OpenProteinSet [1] for training MSAs. We use the January 2023 snapshot of the PDB and a training cutoff of May 1, 2018 and May 1, 2020 for AlphaFold2 and ESMFold. The models are respectively tuned for 1.28M and 720k training examples on the PDB with 40% sequence clustering, crops of size 256, a batch size of 64, and no templates. Training and inference (including the untuned models for comparison) are done without recycling. We use 10 inference steps in all experiments.

### 3.1   PDB Conformational States

We first validate the ability of AlphaFlow and ESMFlow to sample distinct conformational states of proteins deposited in the Protein Data Bank (PDB). We collect all proteins which (1) are represented by 2–30 chains deposited after the AlphaFlow training cutoff, (2) have lengths between 256–768

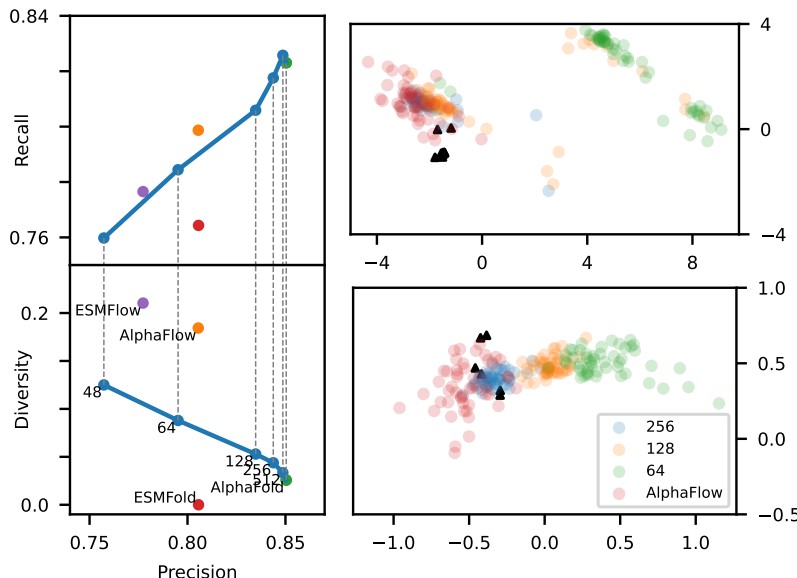

Figure 2: **Comparison of AlphaFlow vs MSA subsampling on PDB conformational states.** *Left*: precision-recall and precision-diversity curves for the benchmarked methods and varying degrees of MSA subsampling. Metrics are reported in terms of symmetrized lDDT-C$\alpha$. *Right*: PCAs of two example targets, showing the experimental structures (▲) along with samples from AlphaFlow and MSA subsampling. Units are in Å RMSD.

residues, (3) have at least two structural clusters when the chains are clustered with a threshold of 0.85 symmetrized lDDT-C$\alpha$ and complete linkage. From the resulting 563 proteins (represented by 2843 chains), we subsample 100 proteins (represented by 500 chains) to form the test set. For each protein, we sample 50 predictions with AlphaFlow/ESMFlow, unmodified AlphaFold/ESMFold, and varying degrees of MSA subsampling using the UniProt reference sequence. Each set of predictions is evaluated on three metrics for each protein: **precision**—the average similarity from a prediction to the closest crystal structure; **recall**—the average similarity from a crystal structure to the closest prediction; and **diversity**—the average dissimilarity between pairs of prediction. Precision and recall are measured with symmetrized lDDT-C$\alpha$ and diversity is measured with $1-$lDDT-C$\alpha$.

The median results across the 100 test targets are shown in Figure 2, *left*. AlphaFlow, similar to MSA subsampling, significantly increases the prediction diversity relative to the unmodified AlphaFold at the cost of reduced precision. Somewhat surprisingly, however, neither AlphaFlow nor MSA subsampling manages to meaningfully improve recall, showing that these methods generally do not succeed in increasing the coverage of experimentally determined structures relative to the baseline AlphaFold. Nonetheless, we highlight that compared to MSA subsampling, AlphaFlow obtains similar recall but significantly higher diversity at the same level of precision. Figure 2, *right* offers an explanation for this phenomenon: because MSA subsampling is an input *ablation* technique, the sampled conformations become more diverse but also drift away from the true structures as the input—and hence predictive signal—is increasingly ablated. In contrast, the AlphaFlow predictions are obtained from the full input and remain clustered around the ground truth conformations while reaching the same or greater levels of diversity.

### 3.2 Molecular Dynamics Ensembles

Unlike inference-time approaches to conformational sampling, our generative modeling framework opens up the possibility of training on ensemble data from beyond the PDB to enable a more comprehensive modeling of protein flexibility. To assess this capability, we tune and evaluate AlphaFlow on protein ensembles from coarse-grained molecular dynamics simulations. Because of the lack of standardized large-scale protein MD data, we construct our own dataset of coarse-grained simulations of 1060 medium-sized proteins (128–256 residues) with no more than 50% sequence similarity. Each protein is simulated with the MARTINI force field [7] and Gō contact restraints [18]

Table 1: **Evaluation on MD ensembles**. For each method, we compare the predicted ensemble with the ground truth MD ensemble according to various metrics (detailed in Appendix B). When applicable, the median across proteins is reported. $\rho$: Spearman correlation; $J$: Jaccard similarity.

| | | AlphaFlow | MSA subsampling | | | AlphaFold |
| --- | --- | --- | --- | --- | --- | --- |
| | | | 16 | 64 | 256 | |
| Predicting flexibility | Ensemble diversity $\rho$ | **0.72** | 0.29 | 0.41 | 0.35 | 0.29 |
| | Global RMSF $\rho$ | **0.81** | 0.35 | 0.52 | 0.55 | 0.52 |
| | Per-target RMSF $\rho$ | **0.86** | 0.50 | 0.62 | 0.66 | 0.66 |
| Distributional accuracy | Atomic $\mathcal{W}_2$-distance | **1.90** | 4.08 | 2.50 | 2.39 | 2.40 |
| | MD PCA $\mathcal{W}_2$-distance | **0.78** | 1.11 | 1.03 | 1.06 | 1.05 |
| | Joint PCA $\mathcal{W}_2$-distance | **1.80** | 4.06 | 2.32 | 2.27 | 2.28 |
| | % PC-sim $> 0.5$ | **25** | 4 | 5 | 3 | 4 |
| Ensemble properties | Weak contacts $J$ | **0.54** | 0.25 | 0.17 | 0.11 | 0.07 |
| | Weak contacts $\rho$ | **0.70** | 0.56 | 0.63 | 0.64 | 0.63 |
| | Exposed residue $J$ | **0.57** | 0.33 | 0.28 | 0.22 | 0.20 |
| | Exposed residue $\rho$ | **0.82** | 0.50 | 0.47 | 0.40 | 0.37 |

for 2.5 $\mu$s with snapshots saved every 100 ps, yielding 25k frames per protein. Training, validation, and test splits of 893/39/99 proteins are defined based on cutoff dates of May 1, 2018 and May 1, 2019 respectively. Further dataset and simulation details in Appendix B. Finally, after all-atom backmapping with GenZProt [25], we fine-tune AlphaFlow (i.e., continuing from the model evaluated on PDB ensembles) for an addition 39k training examples.

To evaluate the models, we sample 250 predictions from AlphaFlow, ESMFlow, and varying degrees of MSA subsampling with AlphaFold for each of the 99 test targets. We then compare these predicted ensembles with to the ground-truth MD ensemble on several metrics of increasing complexity (detailed definitions in Appendix B). We summarize the results (Table 1) as follows:

- Variation in the AlphaFlow ensembles are quantitatively predictive of both global and residue level flexibility (ensemble diversity, RMSF). In contrast, MSA subsampling reveals epistemic uncertainty but does not accurately reflect true physical flexibility.

- The AlphaFlow ensembles are distributionally more accurate, both when the atomic position distributions are considered independently and when considered jointly (projected to the top 2 principal axes). In fact, in 25% of the ensembles the principal component of variation of the AlphaFlow ensemble has $> 0.5$ cosine similarity with the MD principal component.

- MD ensembles are typically intended for downstream analysis of observables such as contacts and solvent exposure [22]. To probe if we model these higher-order properties accurately, we identify the set of *weak contacts* in our predicted ensembles and compare the (1) Jaccard similarity with the ground truth set and (2) the correlation with the ground truth contact probability. We repeat the analysis with the identification of *exposed residues*—those which are buried in the crystal structure but become exposed to solvent in the simulation— and their exposure probability. (Such residues are a key feature in the identification of cryptic pockets [16].) In both cases, AlphaFlow improves significantly over MSA subsampling.

## 4   Conclusion

We have presented AlphaFlow and ESMFlow, which combine AlphaFold2 and ESMFold with flow-matching towards the goal of sampling protein ensembles. We do so via a minimally invasive fine-tuning of existing model weights, retaining the powerful architectures and pretraining investment of these highly-accurate structure predictors. Compared to existing approaches for obtaining multiple structure predictions, our method is the first to go beyond *inference-time* input modifications, is applicable to single-sequence structure predictors such as ESMFold, and takes the first steps towards a more principled *training-time* approach to modeling structural diversity. Hence, our approach represents an important step towards more comprehensive modeling of protein structure and function.

## Acknowledgments

We thank Ruochi Zhang, Hannes Stärk, Soojung Yang, Jason Yim, Umesh Padia, Sergey Ovchinnikov, Gabriele Corso, and Jeremy Wohlwend for helpful feedback and discussions. This work was supported by the NIH NIGMS under grant #1R35GM141861 and a Department of Energy Computational Science Graduate Fellowship.

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

# A  Algorithmic Details

In Algorithms 1 and 2 we provide details of the AlphaFlow / ESMFlow training and inference algorithms, respectively. Note that in the case of ESMFlow, the MSAs are omitted.

---

**Algorithm 1:** TRAINING

---

**Input:** Training examples of structures, sequences, and MSAs $\{(S_i, A_i, M_i)\}$
**foreach** $(S_i, A_i, M_i)$ **do**

> Extract $\mathbf{x}_1 \leftarrow \mathrm{BetaCarbons}(S_i)$ ;
> Sample $\mathbf{x}_0 \sim \mathrm{HarmonicPrior}(L_i)$ where $L_i$ = sequence length of $A_i$ ;
> Align $\mathbf{x}_0 \leftarrow \mathrm{RMSDAlign}(\mathbf{x}_0, \mathbf{x}_1)$ ;
> Sample $t \sim \mathrm{Uniform}[0, 1]$ ;
> Interpolate $\mathbf{x}_t \leftarrow t \cdot \mathbf{x}_1 + (1 - t) \cdot \mathbf{x}_0$ ;
> Predict $\hat{S}_i \leftarrow \mathrm{AlphaFold}(A_i, M_i, \mathbf{x}_t, t)$ ;
> Optimize loss $\mathcal{L} = \mathrm{FAPE}^2(\hat{S}_i, S_i)$ ;

---

**Algorithm 2:** INFERENCE

---

**Input:** Sequence and MSA $A, M$
**Output:** Sampled all-atom structure $\hat{S}$
Sample $\mathbf{x}_0 \sim \mathrm{HarmonicPrior}(L)$ where $L$ = sequence length of $A$ ;
**for** $n \leftarrow 0$ **to** $N - 1$ **do**

> Let $t \leftarrow n/N$ and $s \leftarrow t + 1/N$ ;
> Predict $\hat{S} \leftarrow \mathrm{AlphaFold}(A, M, \mathbf{x}_t, t)$ ;
> **if** $n = N - 1$ **then**
>> **return** $\hat{S}$ ;
>
> Extract $\hat{\mathbf{x}}_1 \leftarrow \mathrm{BetaCarbons}(\hat{S})$ ;
> Align $\mathbf{x}_t \leftarrow \mathrm{RMSDAlign}(\mathbf{x}_t, \hat{\mathbf{x}}_1)$ ;
> Interpolate $\mathbf{x}_s \leftarrow \frac{s-t}{1-t} \cdot \hat{\mathbf{x}}_1 + \frac{1-s}{1-t} \cdot \mathbf{x}_t$ ;

---

Algorithm 3 outlines the architecture of the input embedding module which we attach to AlphaFold2 and ESMFold to form AlphaFlow and ESMFlow, respectively. The output of the input module is then directly added to the input to the Evoformer or folding trunk. The various subroutines are as defined in AlphaFold2 [12]. For brevity, we have omitted droupout layers.

---

**Algorithm 3:** INPUTEMBEDDING

---

**Input:** Beta carbon coordinates $\mathbf{x} \in \mathbb{R}^{N \times 3}$, time $t \in [0, 1]$
**Output:** Input pair embedding $\mathbf{z} \in \mathbb{R}^{N \times N \times 64}$
$\mathbf{z}_{ij} \leftarrow \|\mathbf{x}_i - \mathbf{x}_j\|$ ;
$\mathbf{z}_{ij} \leftarrow \mathrm{Bin}(\mathbf{z}_{ij}, \min = 3.25 \text{ Å}, \max = 50.75 \text{ Å}, N_{\mathrm{bins}} = 39)$ ;
$\mathbf{z}_{ij} \leftarrow \mathrm{Linear}(\mathrm{OneHot}(\mathbf{z}_{ij}))$ ;
**for** $l \leftarrow 1$ **to** $N_{blocks} = 4$ **do**

> $\{\mathbf{z}\}_{ij} \mathrel{+}= \mathrm{TriangleAttentionStartingNode}(\mathbf{z}_{ij}, c = 64, N_{\mathrm{head}} = 4)$ ;
> $\{\mathbf{z}\}_{ij} \mathrel{+}= \mathrm{TriangleAttentionEndingNode}(\mathbf{z}_{ij}, c = 64, N_{\mathrm{head}} = 4))$ ;
> $\{\mathbf{z}\}_{ij} \mathrel{+}= \mathrm{TriangleMultiplicationOutgoing}(\mathbf{z}_{ij}, c = 64)$ ;
> $\{\mathbf{z}\}_{ij} \mathrel{+}= \mathrm{TriangleMultiplicationIncoming}(\mathbf{z}_{ij}, c = 64)$ ;
> $\{\mathbf{z}\}_{ij} \mathrel{+}= \mathrm{PairTransition}(\mathbf{z}_{ij}, n = 2)$ ;

$\mathbf{z}_{ij} \mathrel{+}= \mathrm{Embedding}(t)$ ;

---

## A.1  Comparison with Diffusion

At first glance, flow matching bears a number of similarities with diffusion—both frameworks iteratively generate data from noise, and both permit parameterization with a conditional expectation predictor ($\mathbf{x}_0$ prediction in diffusion, $\mathbf{x}_1$ in flow matching). Hence, it would have been equally valid to train (for example) a harmonic diffusion-based protein generative model [11] with AlphaFold2

or ESMFold as the score model. However, diffusion models rely on the construction of a forward diffusion process which converges to the prior distribution only in the $T \to \infty$ limit, whereas flow matching constructs a probability path which explicitly begins at the prior. In diffusion of protein structure, this difference is crucial as the generative process ought to be *scale-invariant*, i.e., it should not have a bias or reference to proteins of a particular length or spatial size. Prior works with diffusion for protein structure suffered from this limitation: in harmonic diffusion [11], the required diffusion time for convergence directly depends on the protein length, whereas in the VPSDE process (employed for translations in SE(3) diffusion [23, 26]), the variance term is a reference to an arbitrary spatial size. In contrast, flow matching allows us to use a harmonic *prior*—which is scale-invariant—without relying on harmonic *diffusion*—which is *not*—to converge to that prior.

## B  Experimental Details

**PDB test set**  To construct the test set for PDB conformational states, we follow [9] and identify chains as representing the same protein if they map to the same segment in the same UniProt reference sequence. We use the SIFTS annotations database [6] and its residue-level mappings from PDB chains to UniProt reference sequences to associate each deposited chain with a segment. Then, we cluster all segments with a Jaccard similarity threshold of 0.75 and complete linkage, with each resulting cluster regarded as a distinct protein. We then cluster the chains for each protein and select the targets for analysis as further described in the main text.

**Proteins for MD simulation**  To select the proteins for simulation, we collect all chains from the PDB up to the January 2023 snapshot that are between 128–256 residues in length and are the only chain in the respective PDB entry (a proxy for monomeric state). We cluster these 22033 chains at 50% similarity, yielding 6608 clusters. The representative entries are selected if they are non-NMR structures and model a contiguous polypeptide at least 90% of the length of the SEQRES entry. This yields a total of 1060 protein chains for simulation.

**Simulation protocols**  All proteins are simulated in GROMACS [21] with the MARTINI coarse-grained force field [7] and Gō contact restraints, which have been previously shown to faithfully model protein conformational changes [18]. We use explicit MARTINI coarse-grained solvent, 12 Å cutoff for Lennard-Jones and electrostatic interactions, and an integration timestep of 10 fs. Coarse-grained structures undergo energy minimization in vaccum for 400k steps in with steepest-descent, are solvated and neutralized with Na+ and Cl-, and undergo a further 100k steps of minimization. The structure is then heated to 310K in the NVT ensemble with positional restraints and velocity rescaling thermostat for 2 ns. Production simulations are carried out in the NPT ensemble for 2.5 $\mu$s with velocity rescaling thermostat and C-rescaling barostat, with snapshots saved every 100 ps, yielding 25k frames per protein.

**MD evaluation metrics**  To compare a generated ensemble with the ground-truth MD ensemble, we first align all MD snapshots and predicted structures to the PDB structure which initialized the simulation. All analyses are taken at the C$\alpha$ level, i.e., we do not use the MARTINI sidechain beads or the predicted sidechain atoms. We compute the evaluation metrics in Table X, defined as follows:

- *Ensemble diversity*—the Spearman correlation between the expected pairwise RMSDs in the predicted and true ensembles.
- *Global RMSF*—the Spearman correlation between the RMSFs of individual C$\alpha$ in the predicted and true ensembles, pooled over all residues in all test ensembles.
- *Per-target RMSF*—the Spearman correlation between the RMSFs in the predicted and true ensembles; median taken over targets.
- *Atomic $\mathcal{W}_2$ distance.* Because atoms do not have a well-defined position in a structural ensemble, metrics like RMSD which are used to assess single-structure prediction accuracy are not suitable. We thus generalize RMSD by (1) fitting a 3D Gaussian to the positional distribution of each C$\alpha$ and (2) computing the $\mathcal{W}_2$ distance between the corresponding Gaussians in the predicted and ground truth ensembles:

$$\mathcal{W}_2^2(\mathcal{N}(\mu_1, \Sigma_1), \mathcal{N}(\mu_2, \Sigma_2)) = \|\mu_1 - \mu_2\|^2 + \text{Tr}\left(\Sigma_1 + \Sigma_2 - 2(\Sigma_1 \Sigma_2)^{1/2}\right) \quad (7)$$

Note that the squared metric decomposes into a translation component and a variance mismatch component, and reduces to Euclidean distance in the case of point masses. We take the root mean $\mathcal{W}_2^2$ over all C$\alpha$ in a protein and report the median over proteins.

- *PCA $\mathcal{W}_2$ distance.* The atomic $\mathcal{W}_2$ distance does not capture the joint distribution or collective motion of atoms in the ensemble; hence, we perform PCA over the joint distribution of all $3n$ atomic coordinates and compute the $\mathcal{W}_2$ distance between the predicted and true ensembles in this space. (Thermal fluctuations dominate in the full dimensional space and make the $\mathcal{W}_2$ metric unsuitable without an extremely large number of samples.) We take the top 2 principal components computed from (1) the reference ensemble (2) the equally-weighted pooling of the reference and predicted ensembles. While the reference PCA is more commonly used, we note that it can obscure deviations of the predicted ensemble along the orthogonal degrees of freedom.

- *Weak contacts.* Using a 8 Å contact threshold, we define a *weak contact* as a pair of residues in contact more than 1% but less than 99% of the time. We compute the Jaccard similarity between the sets of such contacts in the predicted and true ensembles; and, for all the contacts in their union, the Spearman correlation coefficient between true and predicted contact probabilities.

- *Exposed residues.* Analyses based on solvent-accessible surface area (SASA) are commonly used for all-atom simulations; for evaluative purposes, we apply the same concepts to coarse-grained simulations. For each structure, we compute the SASA of each C$\alpha$ using the Shrake-Rupley algorithm, a probe radius of 2.8 Å and an atomic radius of 2.6 Å. The median SASA in the crystal structure is used as a threshold between buried and exposed residues. For each ensemble, the set of "exposed residues" is then defined to be those which are buried in the crystal structure but have exposure probability greater than 10% in the ensemble. We compute the Jaccard similarity between these sets of residues in the predicted and true ensembles; and, for all the residues in their union, the Spearman correlation coefficient between true and predicted contact probabilities.

