# OpenReview forum: "AlphaFold Meets Flow Matching for Generating Protein Ensembles"
_NeurIPS.cc/2023/Workshop/AI4Science — NeurIPS2023-AI4Science Poster_

### Official Review · Reviewer_bTML · 2023-10-08

**Rating:** 6
**Confidence:** 4

**Review:**

In this work, the authors present fine-tuning regression protein structure prediction model (i.e. AlphaFold2, ESMFold) through flow-matching generative model. The paper demonstrates an valuable direction of using flow-matching for protein conformer ensemble generation and show interesting preliminary results.

Pro:

1. It's a good idea of using flow-matching to generate the ensemble of proteins with different conformations.
2. Fine-tuning pre-trained AlphaFold2 and ESMFold can help make use of existing models and avoid replicating efforts.
3. Modification of MSE loss to consider the invariance in conformer generation.
4. Experiments on functional states, ligand-bind proteins, and disorder are interesting and reflect the drawbacks of previous regression protein structure prediction models.

Cons:
1. Current experiments only include limited number of test proteins.
2. The improvement (e.g., table 1) is marginal when compared with AF2 or ESMFold.
3. It may also worth exploring the benefits of fine-tuning over training from scratch.

---

### Official Review · Reviewer_WnJX · 2023-10-21
**Interesting Take on Protein Ensemble Generation**

**Rating:** 7
**Confidence:** 4

**Review:**

1. Overall evaluation
Opinion:
I believe this paper provides an interesting new route for investigating protein ensembles via flow matching. The conceptual idea is simple yet clever and well-executed. The experiments on functional states and apo/holo proteins are interesting, while the results from the disorder experiments seem less convincing. Overall, I think this paper tackles an important problem in a novel way and is therefore of significant interest to the community. I therefore vote accept.

Summary:
The authors propose a new method to generate protein functional ensembles based on sequence information by harnessing recently developed methods for protein structure prediction. While this idea has previously been explored by input perturbations such as MSA corruption/subsampling and other strategies, the authors instead use the full input information and generate functional state diversity via applying a flow matching objective to the problem. They demonstrate their approach on three different case studies.

Contributions:
[C1] Formulate in the flow matching setting the idea that regression models like structure prediction methods can be used as generative models via this formulation.
[C2] Validate their ideas on three different case studies.

2. Strengths & Weaknesses:
[S1] Novelty of the approach: the authors are the first to use the flow matching objective for protein ensemble sampling and motivate their approach well.
[S2] Strong results in the functional state recovery task: the model indeed seems to be able to generate different functional states accurately.

[W1] Unconvincing disorder section: their task of predicting disorder is interesting, but is only supported by visual inspection of contact maps, which are not very convincing. A more quantitative measure/broader evaluation would have been illuminating.
[W2] No distribution evaluation: while the case studies presented for individual proteins in the different cases illustrate the point of the authors, more results on diverse proteins or benchmarks like Table 1 would have been useful to avoid the possibility of cherry-picking results.

---

### Meta-Review · Area_Chair_9RAX · 2023-10-26

**Recommendation:** Accept (Poster)
**Confidence:** 4

**Metareview:**

This paper proposes to explore protein ensembles via flow matching. The reviewers agree that this is an important problem and the approach is interesting. While the results are limited to a number of proteins, this work should be of interest to the AI4Science community. Recommendation: Poster.